# A Lyapunov Condition for Training ODEs

## Abstract

Control theory is widely used in the study of differential equations to obtain desired behavior from underlying dynamics. We propose a novel method for training ordinary differential equations by using a control-theoretic Lyapunov Condition for stability. This method avoids rolling-out ODE's during training and thus saves the cost of back propagating through a solver or using the adjoint method. We validate our approach experimentally and verify that it has similar performance to ODEs trained by backpropagating through rollouts.

## 1 Introduction

We begin by reviewing how to use ordinary differential equations (ODEs) as a learnable component and some related background in control theory.

### 1.1 Ordinary Differential Equations as Learnable Components

As originally presented in [6, 7], we are concerned with learning a map $x \to y$ using functions $\phi(\cdot; \theta_\phi)$, $\psi(\cdot; \theta_\psi)$ and $f(\cdot, \cdot, \theta_f)$. so that they satisfy the following:

$$h(t_0) = \phi(x; \theta_\phi) \tag{1}$$

$$\frac{dh}{dt} = f(h, t; \theta_f) \tag{2}$$

$$y = \psi(h(T); \theta_\psi) \tag{3}$$

Where, without loss of generality, we assume integration in the time interval $[t_0, T]$. In this formulation, computing gradients with respect to $\theta$ requires rolling out the dynamics and can be done either by backpropagation through a solver or use of the adjoint method.

### 1.2 Lyapunov Conditions for Stability

In a Lyapunov analysis we are primarily concerned with the stability property of a dynamical system:

$$\frac{dh}{dt} = f(h) \tag{4}$$

$$h(t_0) = h_0 \tag{5}$$

Although Lyapunov Theory applies generally to time-varying systems, we will focus on autonomous systems for ease of exposition. The statement of Lyapunov is as follows from [9]:

Submitted to 35th Conference on Neural Information Processing Systems (NeurIPS 2021). Do not distribute.

**Theorem 1.** *Consider the ODE in Equation* (5). *Let* $f : E \to \mathbb{R}^n$ *be a locally Lipschitz continuous function defined on the open and connected set* $E \subseteq \mathbb{R}^n$. *Let* $h^* \in E$ *be an equilibrium point* $(f(h^*) = 0)$ *and* $V_{h^*} : E \to \mathbb{R}$ *be continuously differentiable. If the following conditions hold a system is exponentially stable to* $h^*$.

*$V_{h^*}$ Positive Definite:*

$$V_{h^*}(h) > 0 \quad \text{for all } h \in E \setminus \{h^*\} \tag{6}$$
$$V_{h^*}(h^*) = 0$$

*Local Stability Conditions* $h \in E$:

$$k_1 \|h\|^2 \le V_h^*(h) \le k_2 \|h\|^2 \tag{7}$$
$$\frac{dV_{h^*}}{dt} \le -k_3 \|h\|^2$$

*where* $k_1, k_2, k_3 \in \mathbb{R}_{\ge 0}$.

*Exponential stability implies the following rate of convergence* $\|x(t)\| \le (\frac{k_2}{k_1})^{\frac{1}{2}} e^{-\frac{k_3}{2k_2}(t-t_0)}$ *for all* $t \ge t_0$.

Proof of this theorem is outside the scope of this paper. We note that many common functions in Deep Learning satisfy the requirements of the theorem. For example, Convolutions and ReLu layers are locally Lipschitz. Furthermore, we note that Equation (7) is a local state-dependent condition that when satisfied over the entirety of the state space of the dynamical system, guarantees a global property: exponential stability to an equilibrium point $h^*$.

## 1.3 Contribution

We frame the learning problem in a framework amenable to Control Lyapunov Analysis. This includes showing that various types of loss functions satisfy the requirements of a positive definite function and providing an equivalent reformulation of Equation (3) as an inverse control problem. We then introduce a training procedure that minimizes the violation of the Lyapunov Conditions in expectation. Finally, we present an empirical evaluation of our method that results in a 4x speed up during training while maintaining similar performance with ODE models trained with traditional methods.

## 2 Method

### 2.1 Loss Functions as Lyapunov Functions

Consider the output of a prediction model $z(x; \theta)$. When training a model we typically consider the optimization problem $\operatorname{argmin}_\theta \mathcal{L}(z(x; \theta), y)$ for some loss function $\mathcal{L}$. Possible loss functions include squared error $\mathcal{L}(z, y) = \|z - y\|_2^2$ or cross entropy $\mathcal{L}(z, y) = -\log\left(\frac{\exp z_y}{\sum_i \exp z_i}\right)$. In either case, given a label $y$ the function $\mathcal{L}$ is convex on input $z$ which allows to conclude that the loss function satisfies the positive definite condition of Lyapunov Functions so that $\mathcal{L}(z(x; \theta), y) = V_y(z(x; \theta))$. If $z$ is the output of a dynamical system, we can consider $\mathcal{L}$ as a Lyapunov Function with an equilibrium point at the correct label for $x$.

### 2.2 Supervised Learning as Inverse Control

In a similar fashion to Equation (3), we will attempt to learn the parameters of an underlying dynamical system:

$$h(t_0) = h_0$$
$$\frac{dh}{dt} = f(h, \phi(x; \theta_\phi); \theta_f) \tag{8}$$
$$z = \psi(h(T); \theta_\psi)$$

We can thus interpret the supervised learning problem as finding the parameters $\theta$ that render the input $x$ stable to the label $y$. This can be achieved by satisfying the following condition:

$$\text{for all } h \in E \quad \dot{V}_y(z) = \frac{\partial \mathcal{L}}{\partial z} \frac{\partial \psi}{\partial h} f(h, \phi(x, \phi(x; \theta_\phi)); \theta_f) \leq -\sigma V_y(z) \tag{9}$$

### 2.3 Monte Carlo Method for Inverse Control

The above condition must apply over the all the dynamics state space to guarantee exponential convergence. We can alternatively express Equation (9) as the state integral:

$$\int_E \max \left\{ 0, \dot{V}_y(z) + \sigma V_y(z) \right\} dz \tag{10}$$

For each sample in our dataset we can then approximate this integral through Monte Carlo integration by sampling sates using a uniform distribution over $E$:

$$\mathbb{E}_{z \sim \mathcal{U}(E)} \left[ \max \left\{ 0, \dot{V}_y(z) + \sigma V_y(z) \right\} \right] \tag{11}$$

We can thus summarize the proposed lyapunov learning method as shown in Figure 1.

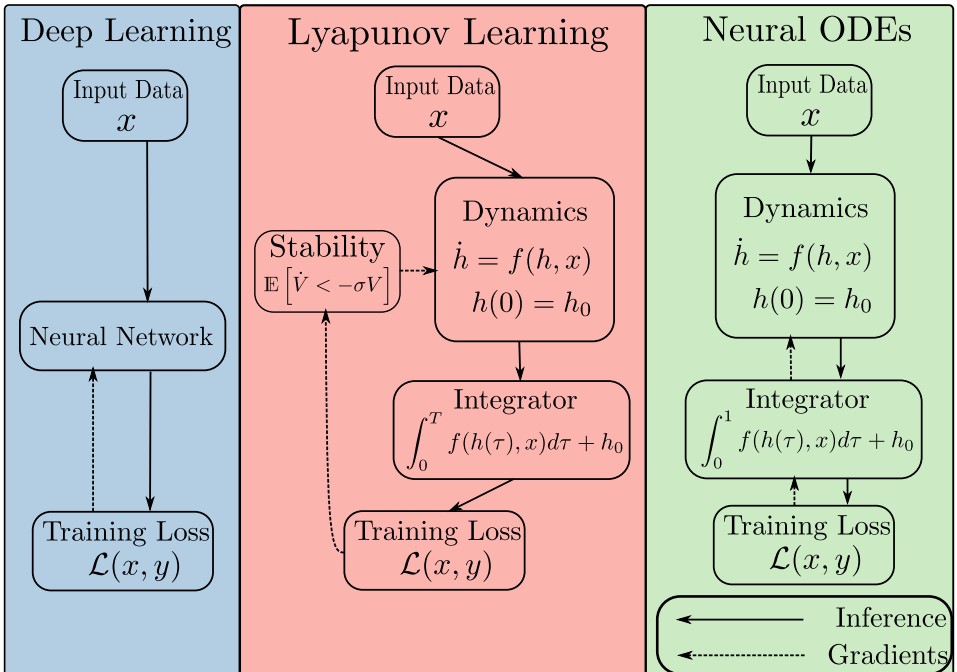

Figure 1: On the left the information flow diagram for Deep Learning where inference flows from the input data to the neural network and the loss. In Neural ODEs, the adjoint method is used to differentiate through the integrator to update the dynamics. Finally, in the Lyapunov learning technique presented here, the stability control theoretic condition allows us to bypass the need to differentiate through the integrator.

## 3 Experiments

We use this Lyapunov method on the MNIST [11] and CIFAR-10 [10] Datasets and compare it with the Adjoint method as presented in [6] as well as AlexNet. The state space of the dynamics satisfies $h \in \mathbb{R}^{10}$ for the following experiments.

|  | LyaNet | Neural ODE | AlexNet |
|---|---|---|---|
| MNIST Mean Test Error | 0.92% | 0.72% | 0.93% |
| MNIST Std. Dev. Test Error | 0.13% | 0.09% | 0.23% |
| CIFAR-10 Mean Test Error | 29.13% | 28.81% | 31.55% |
| CIFAR-10 Std. Dev. Test Error | 0.98% | 1.00% | 0.47% |
| Number of Parameters (1e3) | 52 | 52 | 57,000 |
| Training Time (seconds/epoch) | 77.70 | 316.78 | 12.45 |

Figure 2: For both MNIST and CIFAR-10, AlexNet, NeuralOde and LyaNet (the Lyapunov Learning Method) are compared. For each model the number of parameters and the number seconds per epoch on average across all datasets are included. 5 random seeds were used to run these experiments.

The experiments in Figure 2 were run with a batch size of 128 and learning rate of 0.001 optimized with Adam. These experiments with run on an a system with a single NVIDIA Titan X GPU. We did not perform any tuning to choose these hyper-parameters. Overall we observe similar performance on the ODE-based models that is comparable with the results obtained with AlexNet.

## 4 Related Work

**Prior Work in Learning Dynamical Systems:** Most prior work has focused on using the adjoint method to infer dynamics [6, 2] . The proposal by [7] even discusses properties like controllability but ultimately frames inference as an optimal control problem. Although optimal control is a powerful framework, this representational power comes at cost the cost of fragile solutions and weak guarantees. Alternative approaches have used a similar dynamical system representation in combination with the implicit function theorem to learn a Lyapunov function with its equilibrium point, learn equilibrium networks and even stable equilibrium networks in a similar classification setting[12, 4, 3]. Also, [8] learns stable-by-construction networks that learn a negative-definite decomposition. Still these approaches fail to exploit the Lyapunov-like properties of the loss function as proposed here.

**Prior Work in Learning and Control:** Prior work at the intersection of learning theory and control has focused on using results from one field in the other. For example, [16] use Lyapunov theory to analyze the dynamical system implicit in the momentum updates of stochastic gradient descent, [1, 15] differentiate through controllers like MPC and [13] learn control policies directly for a real system. [14] safely learn physical dynamics by taking into account lyapunov-like conditons during training.[5] use an adversarial approach to learn Lyapunov functions for control.

## 5 Further Work

Future work will focus on scaling this methodology to work with larger networks that have a larger dynamics state. During training we noted that that networks with large dynamics states, in the order of what ResNet uses, would overfit. We also wish to further explore the theoretical properties that stability confers to the learned model in terms of Generalization and Adversarial Robustness.

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
