# OpenReview forum: "A Lyapunov Condition for Training ODEs"
_NeurIPS.cc/2021/Workshop/DLDE — Reject_

### Official Review · Reviewer_chFA · 2021-09-30
**Review for paper number 13 for DLDE Workshop -- NeurIPS 2021**

**Confidence:** 3

**Review:**

The paper presents a new method for training ODEs using a control-theoretic Lyapunov Condition for stability. This method saves the cost of backpropagating through a solver or using the adjoint method. The effectiveness of the method is demonstrated on some benchmark datasets.

Overall the paper is quick demonstration of this new method and is suitable for this workshop. The authors do plan to investigate if this new method scales to larger networks.

Typo on Page 4 - "...at cost the cost...."

**Score:**

3: Good paper

---

### Official Review · Reviewer_XX27 · 2021-10-04

**Confidence:** 4

**Review:**

## Summary

The authors appear to present an idea that a neural ODE can be trained via a stability criterion on the dynamical system. The paper is not very well written, unfortunately: I do not find it clear what the technical idea being communicated is.

As best I can determine, the idea is to train a neural ODE by imposing a training criterion not at the final point of its evolution, but at all points of its evolution.

Overall I would say that this seems like an interesting idea that may well be deserving of further exploration. My low score is primarily a reflection of the difficult presentation.

## Problems

The statement of Theorem 1 could do with being slightly more precise. For example "$\frac{\mathrm{d} V_{h^*}}{\mathrm{d} t}$" should really be "$\frac{\mathrm{d} V_{h^*}(h(t))}{\mathrm{d} t}$". The derivatives of just $V_{h^*}$ alone mean very little; it is specifically the derivative of $V_{h^*}\circ h$ that is of interest.

That $V_{h^*}$ is a Lyapunov function is not stated. Either do or don't assume that the reader is familiar with Lyapunov theory -- if the reader is assumed to be familiar than the introduction is needlessly wordy. If the reader is not assumed to be familiar, then the statements in the paper need to be joined up more carefully.

I am not completely convinced by the characterisation of loss functions as Lyapunov functions. A Lyapunov function says something about the behaviour of a dynamical system (the manner in which $h$ evolves is intrinsically part of its definition), but this is not exhibited by a loss function.

(Addendum to the above: a loss function is a Lyapunov function for the dynamics of *gradient descent*, which is unrelated to the dynamics of the *neural ODE*.)

Generally speaking, I would strongly encourage the authors to explicit denote at what a derivative is evaluated. For example equation (9) is very unclear: at what point is $\frac{\partial \mathcal{L}}{\partial z}$ evaluated?

Typos:

- Equation (7) has $V_h^*$ rather than $V_{h^*}$
- Equations (8) seems to have a $\phi(x;\theta_\phi)$ where originally there was a $t$. This seems to be a mistake. (If it is actually what is meant then I do not know what this meant to denote.) Equation (9) makes things worse and has an unmatched bracket.
- Sentence below equation (11): "lyapunov" is uncapitalised.

## Possible extensions

Sampling $z$ uniformly (equation (11)) seems like it might be expensive/wasteful/ineffective when treating high dimensional datasets. Perhaps $z$ could be sampled from those values obtained during the forward pass of a model.

It is not clear to what extent this applies to neural differential equations more broadly, in contexts other than considering a neural ODE as an initial-to-terminal map. (See for example Section 1.2 of https://arxiv.org/abs/2009.09457, who describe neural differential equations as having primarily other applications. Indeed simple image classification is likely better handled via a traditional residual network or similar.)

## Related work

This work seems somewhat reminiscent of https://arxiv.org/abs/2003.08063, in which the RHS of a neural ODE is characterised as  the gradient of a potential function, i.e. $f_\theta(t, y) = -\nabla_y \phi(t, y)$. By construction the potential $\phi$ is a Lyapunov function.

**Score:**

2: Borderline paper

---

### Official Review · Reviewer_jX45 · 2021-10-05
**Review for paper number 13 for DLDE Workshop -- NeurIPS 2021**

**Confidence:** 2

**Review:**

# Summary
The authors propose a new method to train ODEs using a "control-theoretic Lyapunov Condition for stability". The authors claim that the training avoids the cost of backpropagation through a solver, which sounds promising.

# Problems
The paper is, unfortunately, not written well enough for me to understand the details. Here are a few remarks you can use to improve the paper.

In line 12: ... a map $h: x\to y$

In line 13: semi-colon missing in f.

In line 16: Please add a reference for the adjoint method.

In line 30: Please add: "We refer the reader to a proof of Theorem 1 in~\cite{...}"

In line 26: In the middle of the inequality you should have $V_{h^*}(h)$. The * symbol is not placed correctly.

In line 33: You mention global convergence properties based on local conditions. Do these local state-dependent conditions apply in practice?

In line 56: In equation 9, the parentheses are not balanced.

In line 56: Can you provide some intuition behind this inequality? It seems to come out of the blue and I cannot understand quickly what it is supposed to mean. Can you relate it to theorem 1?

In line 62: The figure provides some intuition regarding what you're doing.

In line 66: This is a table, not a figure.

In line 66: Can you use $\pm$ in the table for the standard deviation instead of having specific rows for it? That is, first row and first column would be $0.92 \pm 0.13\%$.

In line 75: "at cost the cost of" (<- fix this)

In line 80: "...fail to exploit..." -> "...do not explicitly exploit..." (I do not think it was the intention of the authors of these papers to exploit Lyapunov-like properties as you claim.

Lines 81 to 86: I do not understand the purpose of this section. How does this work relate to what you have done?

In line 89: that that

# Possible extensions
1. What happens if you combine your method with the adjoint method? Do you get a speed up as well or a better result?
2. Can you start training with your method and then switch to the adjoint method? Would that give you better results?

**Score:**

2: Borderline paper

---

### Official Review · Reviewer_SLcK · 2021-10-11
**Traning of a neural ODE using Lyapunov stability condition; interesting idea, but not yer well written!**

**Confidence:** 3

**Review:**

To the best of my understanding, the paper proposes the concept of training a neural ODE using the Lyapunov stability condition, and argues that it eliminates backpropogation costs. The current draft, However, is not well-written and incapable of properly conveying both the main idea and the details.

Here I mention a few points and typos:
- In general, main theorem and equation statements should be more clear and accurate, without presuming that general readers, particularly in ML venues, are familiar with the Lyapunov condition.
- I believe it should be $V_{h^{\star}}$, rather than $V_{h}^{\star}$ in the inequality of line 26.
- Imbalanced parenthesis in equation 9 of line 56.
- "lyapunov" should be capitalized in line 61.
- "Figure 2" is actually a table, as far as I'm concerned.
- "At cost the cost" in line 75.
- Duplicated "that" in line 89.

In general, I had trouble understanding the authors' intention in several sections, and I believe mathematical statements should be made clearer and more concrete as well. Although the paper concept is intriguing, the paper itself requires further refining before it can be presented anywhere; I'm interested in seeing future iterations of the paper.

**Score:**

2: Borderline paper

---

### Decision · Program_Chairs · 2021-10-16

**Decision:**

Reject

**Comment:**

The reviews for this paper were distinctly mixed. A recurring theme was the quality of presentation, which in many cases hindered understanding. I would particularly highlight the "Addendum to the above..." sentence of Reviewer XX27; it is not clear that the mathematics have actually been handled correctly.

As such I am afraid this paper is rejected.

On a more positive note, the reviewers have provided a wealth of actionable feedback. I would encourage the authors to take the overall submission experience positively, as I think the breadth and quality of the feedback will allow the authors to dramatically improve their paper in time for the next conference submission.